# The Pivotal Role of Oleuropein in the Anti-Diabetic Action of the Mediterranean Diet: A Concise Review

**DOI:** 10.3390/pharmaceutics14010040

**Published:** 2021-12-25

**Authors:** Andrea Da Porto, Gabriele Brosolo, Viviana Casarsa, Luca Bulfone, Laura Scandolin, Cristiana Catena, Leonardo A. Sechi

**Affiliations:** Clinica Medica, Department of Medicine, University of Udine, 33100 Udine, Italy; gabriele.brosolo@uniud.it (G.B.); viviana.casarsa@gmail.com (V.C.); luca.bulfone1@gmail.com (L.B.); scandolin.laura@spes.it (L.S.); cristiana.catena@uniud.it (C.C.); leonardo.sechi@uniud.it (L.A.S.)

**Keywords:** oleuropein, Mediterranean diet, diabetes, metabolic syndrome

## Abstract

Type 2 diabetes currently accounts for more than 90% of all diabetic patients. Lifestyle interventions and notably dietary modifications are one of the mainstays for the prevention and treatment of type 2 diabetes. In this context, the Mediterranean diet with its elevated content of phytonutrients has been demonstrated to effectively improve glucose homeostasis. Oleuropein is the most abundant polyphenolic compound contained in extra-virgin olive oil and might account for some of the anti-diabetic actions of the Mediterranean diet. With the aim to provide an overview of the possible contributions of oleuropein to glucose metabolism, we conducted a PubMed/Medline search in order to provide an update to the available evidence regarding this interesting compound. This narrative review summarizes the data that was obtained in in vitro and animal studies and the results of clinical investigations. Preclinical studies indicate that oleuropein improves glucose transport, increases insulin sensitivity, and facilitates insulin secretion by pancreatic β-cells, thereby supporting the hypothesis of the possible benefits of the control of hyperglycemia. However, on the clinical side, the available evidence is still preliminary and requires more extensive investigations. Thus, many questions remain unanswered in regards to the potential benefits of oleuropein in diabetes prevention and treatment. These questions should be addressed in appropriately designed studies in the future.

## 1. Introduction

It is now more than 50 years since the benefits of the Mediterranean diet on the prevention of cardiovascular and neurodegenerative diseases, cancers, and diet-related metabolic conditions, like diabetes, have been convincingly demonstrated [1]. Diabetes is a major health problem in economically developed countries and is one of the leading causes of morbidity and mortality worldwide. Type 2 diabetes is the most common form of disease and can be prevented, mainly through lifestyle interventions, notably dietary modifications. In fact, the reduced incidence of type 2 diabetes has been reported in prospective studies, which have demonstrated the important benefits of the Mediterranean diet [2,3]. Dietary interventions are also one of the mainstays of the treatment of type 2 diabetes and large studies have consistently demonstrated that these interventions, including the Mediterranean diet, have significant impacts on the clinical outcomes of this condition [4]. Evidence of the benefits of these dietary interventions has therefore triggered a wide interest in the role of functional foods and the bioactive components of the Mediterranean diet for diabetes prevention and treatment.

Substantial consumption of extra-virgin olive oil (EVOO), vegetables, legumes, fruits, and whole grain cereals is the distinctive feature of the Mediterranean diet that replicates some of the traditional eating habits of the countries bordering the Mediterranean Sea. Since the time of Hippocrates of Kos who named it “The Great Healer”, EVOO has shown medical properties that have been held in high regard by countless generations of physicians [5]. Nowadays, the biological effects of EVOO have received broad recognition in the clinical setting and many dietary intervention studies have consistently reported substantial benefits on human health [6]. Along with the Mediterranean diet, which has continuously proven to be a valuable contributor to the dietary treatment of diabetic patients [7], benefits of regular consumption of EVOO have been reported in the management of type 2 diabetes [8]. The effects on the glucose metabolism of specific components of EVOO, however, are still debated.

The medical proprieties of EVOO can be ascribed to some of its chemical components, including fatty acids (triacylglycerols, free fatty acids, mono and diacylglycerols) and other lipids (hydrocarbons, sterols, aliphatic alcohols, and tocopherols), and volatile compounds, including polyphenols, which develop in response to insect injuries in olive trees. The main sources of olive polyphenols are olive leaves that possess the highest antioxidant and scavenging power among the different parts of the olive tree. There are five different chemical groups of phenolic compounds in olive leaves: oleuropeoides, flavones, flavonols, flavan-3-ols, and substituted phenols. Polyphenolic compounds are particularly abundant in EVOO, and multiple beneficial biological effects of these compounds have been reported, including antioxidant, anti-inflammatory, antiproliferative, and finally, antiobesity and antidiabetic properties [9,10,11].

In recent years, oleuropein has developed a growing interest in regard to the phenolic compounds contained in EVOO because of its broad metabolic effects. These effects might contribute to the benefits that are commonly seen in subjects who eat a Mediterranean diet. This narrative article has been written with the purpose of highlighting the possible metabolic benefits of oleuropein, with a specific focus on its effects on glucose metabolism and diabetes. Information was obtained by a PubMed/MEDLINE systematic search of articles that included oleuropein and diabetes as key words. Experimental data obtained in vitro and in animal studies and the results of clinical investigations are included and overviewed. This article will provide the readership with an update on the currently available evidence on this interesting compound and hopefully trigger further clinical research in order to develop better knowledge and to set the stage for its routine use.

## 2. Oleuropein and Diabetes: Experimental Studies

Oleuropein is an iridoid molecule of oleuropeoide and is esterified with a dihydrocaffeoyl alcohol residue that provides the phenolic characteristic to this molecule. It is present in EVOO, both in a glycated and aglycone (iridoid esterified with the phenylpropanoid alcohol) (Figure 1) form, and is the most abundant among the polyphenolic compounds of olive leaves. Like other phenolic compounds contained in EVOO, oleuropein exhibits antioxidant [12], anti-inflammatory [13], and antiproliferative [14] properties and exerts protective actions at the cardiovascular [15], metabolic [12,15], neurological [16], and hepatic [17] level (Table 1).

Moreover, oleuropein has already demonstrated interesting blood glucose lowering properties [18]. The results of experimental studies suggest that the hypoglycemic effects of oleuropein could be mediated by modulation of multiple intracellular signaling mechanisms that are directly involved in the regulation of blood glucose concentration [12]. Exposure of cultured mouse myoblasts to physiological concentrations of oleuropein promoted translocation of the glucose transporter GLUT-4 (glucose transporter-4) to the plasma membrane. This effect was mediated by the activation of adenosine monophosphate-activated protein kinase (MAPK) and was associated with the increased cellular internalization of glucose [19]. The same effect was replicated in vivo in oleuropein-fed mice in whom insulin sensitivity was significantly improved [19]. In vitro studies indicate that the activity of the high-capacity, low-affinity, glucose transporter GLUT-2 (glucose transporter-2) is decreased in cultured cells exposed to oleuropein [20]. GLUT-2 is highly expressed in the gut and is part of the glucose sensor of pancreatic β-cells, thereby playing a central role in the regulation of intestinal glucose absorption and glucose-stimulated insulin secretion. Furthermore, experiments conducted in cultured hepatocytes and in mice exposed to oleuropein (100 mg/kg/day p.o.) for six weeks demonstrate that oleuropein acts as a ligand of the peroxisome proliferator activated-receptor alpha (PPARα), a nuclear receptor that activates a family of genes that are critically involved in many metabolic pathways [21]. PPARα activates gene encoding enzymes that are involved in fatty acid cell uptake, mitochondrial β-oxidation, microsomal **ω**-oxidation, and genes that encode many apolipoproteins [21]. Therefore, PPARα participates in the control of circulating fatty acids with all their related effects, including those on insulin sensitivity and glucose metabolism. Finally, it was reported that oral administration of oleuropein (50 mg/kg/day p.o.) to cholesterol-fed rats for eight weeks decreased body weight and adipose tissue mass and attenuated liver fat deposits. These effects on lipid uptake and adipogenesis were associated with the significant inhibition of peroxisome proliferator activated-receptor gamma (PPARγ) and increase in serum adiponectin levels [22] with important implications for the development of obesity.

In addition to the effects on metabolic pathways and insulin sensitivity, oleuropein facilitates glucose-stimulated insulin secretion by pancreatic β-cells. This experimental evidence was initially reported by Wu et al. who demonstrated a dose-dependent effect of oleuropein in INS-1 (rat insulinoma cell line) β-cells that was linked to activation of the MAPK pathway [23]. Moreover, it was demonstrated that oleuropein prevents cytotoxicity of β-cells that is induced by amyloids, which is a distinctive feature of type 2 diabetes. More recently, it has been demonstrated that oleuropein aglycone inhibits pancreatic β-cell cytotoxicity by human islet amyloid polypeptide aggregates and protects the plasma membrane from permeabilization, thereby preventing cell death [24].

Benefits of oleuropein on blood glucose control have been reported in many studies conducted using animal models of diabetes. Initial evidence was obtained in experiments conducted on rabbits with alloxan-induced diabetes who were treated with oleuropein (25 mg/kg/day p.o.) for 16 weeks. In these animals, oleuropein significantly reduced blood glucose and malondialdehyde, which is a marker of oxidative stress that is markedly increased in this experimental model and might contribute to diabetic complications [25]. Similarly, the administration of oleuropein (8–16 mg/kg/day p.o.) for four weeks reduced blood glucose and improved glucose tolerance in rats with alloxan-induced diabetes [26] and other rodent models of insulin-deficient (streptozotocin-diabetic rats) and insulin-resistant (leptin-receptor deficient ob/ob, Tsumura-Suzuky, fat diet obese mice) diabetes [18]. The effects of oleuropein have also been tested in gestational diabetes using a mice model [27]. In these animals, oleuropein (5–10 mg/day intraperitoneally) attenuated the body weight increase and efficiently decreased blood glucose, plasma insulin, and hepatic glycogen levels, with an overall improvement in the gestational outcome after 20 gestational days. Very recently, Zheng et al. tested the effects of oleuropein (200 mg/kg/day p.o.) administration for 15 weeks in diabetic *db/db* mice, reporting a significant decrease in fasting blood glucose levels, an improvement to glucose tolerance, a lowering of homeostasis model assessment insulin-resistance index, and a promotion of protein kinase B activation [28]. The *db/db* mice model is characterized by pancreatic islet damage, lipid deposition in hepatocytes, and disordered myocardial fibers, which are all histopathological changes that were restored by oleuropein administration. [28]. Additionally, and most importantly, this study examined the composition of the gut microbiota, which shows remarkable changes in the intestinal bacteria in diabetic mice fed with oleuropein. At the phylum level, the relative abundance of *Verrucomicrobia* and *Deferribacteres* was increased by oleuropein, whereas the abundance of *Bacteroidetes* decreased; at the genus level, the relative abundance of *Akkermansia* increased with oleuropein, whereas *Prevotella*, *Odoribacter*, *Ruminococcus*, and *Parabacteroides* decreased [28]. In agreement with the above findings obtained with oleuropein, robust experimental evidence suggests that polyphenols contained in EVOO have important beneficial effects on intestinal bacteria [29]. These findings support the intriguing hypothesis that oleuropein may improve glucose metabolism in diabetes through the modulation of the composition and function of the gut microbiota and open an interesting area of investigation for future studies.

It is well known that gut microbiota has a central role in the maintenance of physiological homeostasis and that its dysregulation (dysbiosis) can modify intestinal permeability, thereby affecting several pathophysiological mechanisms that can have an impact on glucose metabolism. Diverse families of intestinal bacteria mediate their beneficial effects through the fermentation of dietary fibers with the production of short-chain fatty acids. These fatty acids are endogenous signals that play important roles in the regulation of metabolic pathways, including those involved in glucose homeostasis.

Important differences in the doses of oleuropein, used both in acute and chronic experiments, different durations of exposure, and differences in the route of administration to experimental animals do not permit the definition of the optimal doses of this compound, which would be needed to obtain the best metabolic responses.

In summary, data obtained in the animal models of diabetes indicate that oleuropein interferes beneficially with glucose metabolism. These benefits could result from changes in the glucose transport and intracellular metabolism, increased insulin sensitivity, and more efficient glucose-stimulated insulin secretion by pancreatic β-cells (Figure 1).

Oleuropein could regulate the glucose metabolism that acts at many levels and in different organs. On the one hand, oleuropein facilitates insulin secretion by pancreatic β-cells and reduces intestinal glucose absorption. On the other hand, oleuropein improves insulin sensitivity in skeletal muscles and the liver, thereby increasing glucose internalization and decreasing the circulating levels of free fatty acids, thus leading to reduced fat accumulation. The final effects demonstrated in animal studies and suggested by the results of early clinical trials are weight loss and the reduction of fasting and post-oral load glucose. These actions of oleuropein may result in the effective prevention of new-onset diabetes and renal and cardiovascular renal complications. GLUT, glucose transporter; PPAR, peroxisome proliferator activated-receptor; MAPK, adenosine monophosphate-activated protein kinase; AMP, adenosine mono phosphate.

## 3. Oleuropein and Diabetes: Clinical Evidence

As previously stated, the benefits of the regular consumption of EVOO in the management of diabetes have been repeatedly reported. Eleven obese patients with type 2 diabetes, who were treated with oral antidiabetic agents, ate refined oil polyphenol-free for four weeks. This was then replaced by polyphenol-rich EVOO for a further four weeks [30]. Polyphenol-rich EVOO significantly reduced body weight, fasting plasma glucose, and glycated hemoglobin and these changes were associated with a significant decrease in serum visfatin, a pro-inflammatory adipocytokine. Schwingshackl et al. conducted a meta-analysis in order to examine the association between EVOO consumption and the risk of type 2 diabetes, in addition to the effects of EVOO on its management [8]. The analysis included 15,784 subjects who were included in four cohort studies and 29 intervention trials. The results of this meta-analysis demonstrated that the highest EVOO intake category had a 16% lower risk of developing type 2 diabetes, as compared to the lowest. Additionally, in type 2 diabetics, EVOO supplementation resulted in a significantly greater reduction of fasting plasma glucose and glycated hemoglobin than in the control subjects. These studies provided solid evidence that the intake of EVOO could be beneficial for the prevention and management of type 2 diabetes, but did not provide any insight into the contribution of single components of EVOO.

With specific regard to oleuropein, the effects on glucose metabolism and diabetes have been examined in clinical studies and recent data provide initial evidence of the potential benefits (Table 2). In a randomized, double-blind, crossover trial conducted in New Zealand, 46 middle-aged, overweight men received oleuropein containing capsules (51 mg/day) or a placebo for 12 weeks [31]. As compared to the placebo, oleuropein supplementation was associated with a significant improvement in insulin sensitivity and pancreatic β-cells secretory capacity. Kerimi et al. conducted seven separate randomized, crossover, double-blind, placebo controlled, intervention trials on healthy volunteers to examine the effect of oleuropein on post-prandial blood glucose after the consumption of bread, glucose, or sucrose [20]. Oleuropein (35 to 200 mg/day) in solution attenuated the post-prandial blood glucose response after the consumption of sucrose, but did not affect post-prandial glucose after the ingestion of bread or glucose. Examination of the effects of oleuropein on enzymes involved in carbohydrate digestion showed the inhibition of sucrase and GLUT-2-mediated transport, but no significant effect on α-amylase, thus explaining the findings regarding the post-prandial blood glucose changes. Finally, in an open study of hypertensive patients, many of whom had obesity and/or diabetes, oral supplementation of oleuropein (100 mg/day) was administered for two months. In these patients, oleuropein decreased fasting blood glucose, together with other markers of a metabolic syndrome, such as waist circumference and serum triglycerides [32].

Clinical investigations have also tried to clarify some of the mechanisms that might mediate the effects of oleuropein on glucose metabolism. These investigations have brought the possibility that the incretin axis could be modulated by oleuropein to the forefront. The post-prandial glycemic profile was investigated in a crossover study of 25 healthy subjects that were randomly allocated to a Mediterranean diet with or without supplementation of oleuropein (20 mg). Two hours after the meal, subjects who ate oleuropein supplements had a significantly lower blood glucose and dipeptidyl-peptidase 4 (DPP-4) protein concentration and activity, and higher serum insulin and glucacon-like peptide-1 (GLP1) levels [33]. As an extension of this study, the same research group reported that the effects of oleuropein on blood glucose and DPP-4 were associated with a significant reduction in markers of oxidative stress, such as soluble NADPH oxidase-derived peptide activity and 8-iso-prostaglandin-2α [34]. The same protocol with a Mediterranean diet meal with or without oleuropein supplementation was later applied to 30 subjects with impaired fasting glucose [35]. In agreement with the findings obtained in healthy subject, the meal containing oleuropein was associated with a reduction of blood glucose and DPP-4 activity and an increase in insulin and GLP-1, as compared to the meal without oleuropein. Finally, similar experiments were conducted with a crossover design in 25 type 2 diabetic patients who were randomized to receive 40 g of oleuropein in the form of oleuropein-enriched chocolate or control chocolate [36]. In diabetic patients who received oleuropein-enriched (40 mg) chocolate, the increase in blood glucose following ingestion was smaller than in diabetic patients who received plain chocolate, and even in that case, the effect of oleuropein was associated with decreased DPP-4 activity.

In summary, initial clinical observations suggest that there is a potential for oleuropein use in the control of hyperglycemia. The findings suggest that oleuropein might decrease post-prandial blood glucose with a mechanism that counteracts oxidative stress-mediated incretin down-regulation. While more comprehensive evidence is required, the effects of oleuropein might provide both preventive and therapeutic benefits to patients with type 2 diabetes. Due to the significant variability in doses of refined oleuropein that were used in acute and chronic studies, and in the amount of oleuropein contained in EVOO, the current evidence does not allow us to establish suitable doses of this compound for clinical use.

## 4. Oleuropein and Renal Complications: Experimental Evidence

It is important to note that the potential of oleuropein to be beneficial in the context of diabetes is not limited to the control of metabolic balance with the reduction of hyperglycemia, but this compound might also have an impact on diabetic complications. For instance, some studies suggest the possibility that oleuropein may exert protective actions on the kidney, which is one of the main targets of diabetes. These protective actions of oleuropein were recently demonstrated in experimental animal models of renal damage. First, oleuropein (50 mg/kg of body weight) was administered to rats in whom acute kidney injury was induced by a glycerol injection [37]. This model of renal damage is closely linked to the activation of proinflammatory mechanisms and oxidative stress demonstrated by elevated cytokines and malondialdehyde and decreased glutathione content. Furthermore, renal damage is associated with rhabdomyolysis and multiple molecular, biochemical, and histological alterations. All these alterations were reversed by oleuropein, thus suggesting that this compound might have the potential to be a treatment for this condition. Second, oleuropein was tested in rats with a kidney ischemia-reperfusion injury, and even in this case, renal damage was reduced with a significant improvement in creatinine, urea, uric acid, and lactate dehydrogenase levels [38]. In this model, oleuropein administration (10, 50, and 100 mg/kg) decreased proinflammatory (C-reactive protein and expression of cyclooxygenase 2) and pro-apoptotic (caspase-3) markers, and up-regulated antioxidant capacities that were identified by activated catalase, superoxide dismutase, and glutathione peroxidase activity. Third, kidney injury was induced in rats by three-day unilateral ureteral obstruction and oleuropein (50, 100, and 200 mg/kg) was administered in increasing doses, which improved renal response in a dose-dependent way [39]. Even in this model, renal improvement was associated with activated superoxide dismutase and glutathione peroxidase and down-regulation of pro-inflammatory (tumor necrosis factor-α) and pro-apoptotic (caspase-3, Bax protein) molecules. Thus, all these studies linked the observed benefits of oleuropein on renal function to the evidence of important antioxidant, anti-inflammatory, and anti-apoptotic effects. These effects of oleuropein might also come into play in other more chronic renal conditions, including diabetic nephropathy.

In summary, although oleuropein has been beneficial in different experimental models of renal damage, no data is currently available on the use of oleuropein in the context of diabetic nephropathy. This could be an issue worth testing in future animal and human studies.

## 5. Oleuropein and Cardiovascular Complications: Experimental Evidence

Another major issue is related to the possible role of oleuropein in the protection from cardiovascular events that are the principal cause of morbidity, disability, and mortality in patients with type 2 diabetes. Oleuropein (3% as oral supplements) has been shown to reduce body weight gain and abdominal fat in high fat-fed rats with a mechanism that appears to be linked to the repression of mitochondrial activity during the differentiation of adipocytes [40]. Inhibition of clonal expansion with the prevention of differentiation and intracellular triglyceride accumulation was also demonstrated in cultured 3T3-L1 preadipocytes because of the reduced expression of adipogenesis-related genes [41]. Animal studies suggest that oleuropein might favorably affect some risk factors that, together with diabetes, contribute to the global cardiovascular risk [15]. In alloxan-diabetic rats, oral supplementation with oleuropein (8–16 mg/kg/day for 4 weeks) has been shown to reduce total cholesterol, serum LDL, and triglycerides and to increase serum HDL [26]. Oleuropein increases LDL-receptors of hepatocytes and regulates the expression of genes that are involved in metabolism and the disposal of triglycerides [19]. Most of these effects of oleuropein on lipid metabolism appear to be related to increased activation of PPARα, with many of its target genes having a mechanism like that of fibrates [21]. In a rat model of simultaneous diabetes and hypertension, oleuropein was administered at increasing doses (20, 40, and 60 mg/kg/day) for four weeks and was compared to the vehicle [42]. Oleuropein significantly reduced blood pressure, blood glucose, total and LDL cholesterol, and triglycerides, and improved glucose tolerance. These effects were apparently linked to antioxidant and simpatho-inhibitory mechanisms [43], and released nitric oxide, which, at physiological levels, activate the soluble guanylate cyclase signal transduction pathway and thereby leads to various beneficial effects [44]. Moreover, in the same experimental diabetic model (20, 40, and 60 mg/kg/day for 4 weeks), oleuropein reduced infarct size and coronary effluent creatine kinase after cardiac ischemia-reperfusion injury, in comparison to controls, thus supporting the hypothesis of a cardioprotective effect of this compound [45]. Even in this case, the benefit of oleuropein was associated with activation of superoxide dismutase, thus suggesting the involvement of antioxidant mechanisms. Regarding hypertension, in a double-blind, randomized, crossover dietary intervention study, 24 women with high-normal blood pressure or stage 1 hypertension received either polyphenol-rich oil (30 mg/day) or polyphenol-free oil for two months. Only the polyphenol-rich oil led to a significant fall in systolic and diastolic blood pressure, which was associated with significantly decreased serum asymmetric dimethylarginine, thus suggesting improved endothelial functioning [46].

In summary, there is evidence of the possible benefits of oleuropein on overweight/obese, dyslipidemia and hypertension patients that might result in better protection from cardiovascular events in high-risk conditions, such as type 2 diabetes.

## 6. Perspectives

The increase in life expectancy observed in the Western World is paying a price to the progressively greater incidence of many chronic diseases associated with ageing and lifestyle. For this reason, medical research is progressively shifting its focus from a cure to a prevention, with greater relevance attributed to the “lifestyle” concept that primarily involves diet and thus food. Epidemiological evidence has provided support to the idea that “healthy diets”, such as the Mediterranean diet, are associated with a lower incidence of chronic conditions. Therefore, there is now rapidly growing interest in the effects of the bioactive components of food.

An important feature of the Mediterranean diet is the elevated content of phytonutrients that affect multiple biological functions, including glucose homeostasis. Type 2 diabetes is the most common form of diabetes, currently accounting for more than 90% of all diabetic patients. Lifestyle interventions and notably dietary modifications are one of the mainstays of type 2 diabetes prevention and treatment. For this reason, dietary components have been extensively investigated in order to find which of them could sensibly affect glucose metabolism. For instance, the dietary content of advanced glycation end products (AGE) has been investigated and a recent meta-analysis has reported that a poor diet of AGE has beneficial effects on fasting insulin and insulin resistance [47]. However, several bioactive compounds have been reported to favorably affect glucose homeostasis in type 2 diabetes [48]. Polyphenols are highly represented in EVOO, and in turn, it is one of the main components of the Mediterranean diet. Among the polyphenols contained in EVOO, oleuropein has gathered growing interest because of its antioxidant, anti-inflammatory, and antiproliferative properties, which have been demonstrated in experimental in vitro and in vivo studies. With specific regard to glucose metabolism, initial clinical observations suggest that there is a potential for oleuropein use in the prevention and treatment of diabetes.

Oleuropein is only one of a myriad of bioactive compounds of food that deserve careful evaluation by both basic and clinical researchers. Expansion of knowledge on the effects of oleuropein and other food components on human health, together with the refinement of manufacturing for market use, will hopefully permit future, effective preventive and therapeutic interventions to reduce the burden of chronic disease.

## 7. Conclusions

A Mediterranean dietary pattern is effective in managing nutrition-related metabolic disorders, including diabetes. Polyphenols are under the spotlight as they are compounds with greater biological interest among the nutraceuticals contained in the Mediterranean diet. Therefore, the potential of oleuropein, as a major component of EVOO in the Mediterranean diet for modulation of glucose metabolism, is rapidly gaining interest within the scientific community. Preclinical studies indicate that oleuropein improves glucose transport and intracellular metabolism, increases insulin sensitivity, and facilitates insulin secretion by pancreatic β-cells, thus supporting the intriguing hypothesis that this phenolic compound might be beneficial for the prevention of diabetes and control of hyperglycemia. However, the available clinical evidence is too preliminary and must be strengthened in larger and appropriately designed studies. Strengthening the evidence in the clinical field would require some essential steps. First, sources of oleuropein should be carefully defined by standardization of olive leaf extraction and purification procedures in order to have control of the administered amount that, up to now, has been extremely variable. Second, dose finding studies should be planned to define the threshold for oleuropein effects that may differ depending on the outcome of various interests. Third, appropriate assessment of the exposure time that is necessary to obtain biologically relevant effects will be needed. Finally, like any other dietary intervention, it would be useful to have laboratory tests that permit the definition of adherence to oleuropein use in chronic studies.

While many commercial preparations of oleuropein obtained from olive leaves extraction are currently invading the market, more still needs to be learnt regarding the best sources, the appropriate intake, and the exposure time that is necessary to obtain clinical benefits in people with type 2 diabetes or that are at risk of diabetes. Future randomized clinical trials will hopefully assess the evidence needed for the widespread use of oleuropein in daily clinical practice.

## Figures and Tables

**Figure 1 pharmaceutics-14-00040-f001:**
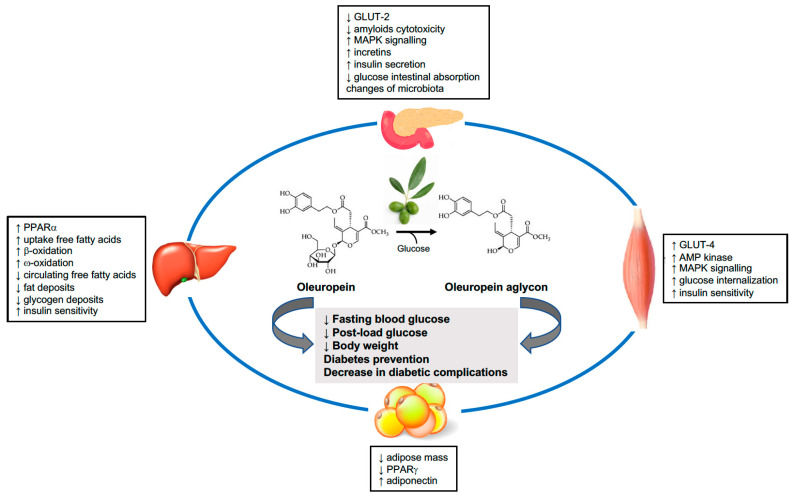
Effects of oleuropein on glucose homeostasis and the related mechanisms.

**Table 1 pharmaceutics-14-00040-t001:** Biological properties and effects of oleuropein.

Properties	Possible Mechanisms
Anti-Oxidation [12]	reactive oxygen species scavenging
	improved free radical stability
	increased catalase, superoxide dismutase, glutathione peroxidase, thioredoxin reductase activity
	decreased malondialdehyde, advanced glycation endproducts
Anti-Inflammation [13]	decreased C-reactive protein, neutral factor-κB, interleukin-1β, interleukin-6, adipocytokines, tumor necrosis factor
	lipooxygenase inhibition
Anti-Cancer [14]	inhibition of cell proliferation, angiogenesis, cell migration
	induction of apoptosis
	reactive oxygen species scavenging
	inhibition of human epidermal growth factor receptor, Bcl-2A pathways, protein kinases, neutral factor-κB, cyclinD1
	activation of Bax, Janus kinase
Cardiovascular Protection [15]	reduced oxidative stress
	increased nitric oxide formation
	lipid lowering and reduced lipid peroxidation
	reduced blood pressure
Metabolic Protection [12,15]	decreased obesity
	reduced blood glucose
	diabetes prevention
Neuroprotection [16]	reduction of oxidative stress
	stabilization of amyloid fibers
Hepatoprotection [17]	reduction of oxidative stress
	reduction of fat deposition

**Table 2 pharmaceutics-14-00040-t002:** Clinical trials that examined the effects of oleuropein on glucose homeostasis.

Reference	Study Design	Source, Main Content and Time of Exposure to Oleuropein	Effects on Glucose Metabolism
Kerimi et al. [17]	RCT, double-blind, crossover 24 healthy volunteers	Supplement vs. Placebo 35–200 mg—Single dose	Reduction of Post-prandial glucose Inhibition of GLUT2 and maltase
De Bock et al. [28]	RCT, double-blind, crossover 46 overweight volunteers	Olive Leaf vs. Placebo 51.1 mg vs. Placebo 12 weeks	Improvement in insulin sensitivity Improvement in pancreatic β-cell responsiveness
Hermans et al. [29]	Prospective, open observational 663 Hypertensive patients	Supplement 100 mg 8 weeks	Reduction of fasting glucose
Violi et al. [30]	RCT, double-blind, crossover 25 healthy volunteers	EVOO vs. Corn Oil 20 mg Single dose	Reduction of post-prandial glucose Inhibition of DPP-4 Improvement in GLP-1 mediated insulin secretion
Carnevale et al. [31]	RCT, double-blind 20 healthy volunteers	EVOO vs. Corn Oil 20 mg Single dose	Reduction of post-prandial glucose Inhibition of DPP-4 Improvement in GLP-1 mediated insulin secretion
Carnevale et al. [32]	RCT, double-blind 30 patients with IGT	EVOO vs. Corn Oil 20 mg Single dose	Reduction of post-prandial glucose Inhibition of DPP-4 activity Improvement in GLP-1 mediated insulin secretion
Del Ben et al. [33]	RCT, single-blind 25 patients with Type 2 Diabetes 20 healthy volunteers	EVOO Enriched vs. Standard Chocolate 40 mg Single dose	Reduction of post-prandial glucose Inhibition of DPP-4 activity Improvement in GLP-1 mediated insulin secretion

EVOO, Extra Virgin Olive Oil; IGT, Impaired Fasting Glucose; DPP-4, Dipeptidlpepdisase-4; GLP-1, Glucagon Like Peptide-1.

## Data Availability

Not applicable.

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
