# Peer review of "The Pivotal Role of Oleuropein in the Anti-Diabetic Action of the Mediterranean Diet: A Concise Review"

_pharmaceutics, 2021, doi:10.3390/pharmaceutics14010040_

Round 1

Reviewer 1 Report

The manuscript in reference describes an interesting description as a commentary regarding the oleuropein as an important anti-diabetic bioactive of the Mediterranean diet. I found this manuscript very interesting and it provides relevant comments of this bioactive component. However, some minor points should be addressed prior acceptance.

  1. A detailed scrutiny throughout the manuscript is required to revise some grammar and stylistic issues to improve the quality of this manuscript.
  2. Lines 52-53: The structural description about oleuropein is a little bit confusing, since an iridoid per se is not a phenolic compound. I consider important to expand the description of this very interesting metabolite by indicating that oleuropein is an iridoid esterified with a dihydrocaffeoyl alcohol residue, which provide the phenolic characteristic to this molecule.
  3. Line 53: Oleuropein is a glycoside. Clarify what is the name of the aglicone (iridoid esterified with the phenylpropanoid alcohol). In addition, I recommend to include an structure to extent, clarify and explain better this structural information about oleuropein for readers.
  4. Line 172: Specify that this section are only related to animal model-based evidences.
  5. Line 195: These ideas should be included into a new section.
  6. A perspective-related section is missing, before conclusions.
  7. Line 233: how can the clinical evidences be strengthened? This ideas (and the subsequent ones) are very laconic. Authors can be more specific for readers.
  8. Revise the association of the manuscript content and the title, since the relationship of the benefits in the Mediterranean diet is only mentioned in the Introduction section. Perhaps in conclusions or perspectives sections something about this can be mentioned.

Author Response

Reviewer 1

We thank the reviewer for his/her thoughtful comments and suggestions. Here are our responses to the points that were raised.

The manuscript in reference describes an interesting description as a commentary regarding the oleuropein as an important anti-diabetic bioactive of the Mediterranean diet. I found this manuscript very interesting, and it provides relevant comments of this bioactive component. However, some minor points should be addressed prior acceptance.

Au. Thank you

  1. A detailed scrutiny throughout the manuscript is required to revise some grammar and stylistic issues to improve the quality of this manuscript.

Au. Grammar and stylistic issues have been revised as requested.

  1. Lines 52-53: The structural description about oleuropein is a little bit confusing, since an iridoid per se is not a phenolic compound. I consider important to expand the description of this very interesting metabolite by indicating that oleuropein is an iridoid esterified with a dihydrocaffeoyl alcohol residue, which provide the phenolic characteristic to this molecule.

Au. The structural description of oleuropein has been modified according to the Reviewer’s suggestion (page 2, lines 78-79). Thank you.

  1. Line 53: Oleuropein is a glycoside. Clarify what is the name of the aglicone (iridoid esterified with the phenylpropanoid alcohol). In addition, I recommend to include an structure to extent, clarify and explain better this structural information about oleuropein for readers.

Au. The structural detail of oleuropein aglycone has been clarified in text as requested (page 2, lines 79-81). Also, structure of oleuropein has been shown in a new figure (Figure 1). Thank you.

  1. Line 172: Specify that this section are only related to animal model-based evidences.

Au. The title of the section has been modified as suggested (page 7, line 239 and page 8, line 273).

  1. Line 195: These ideas should be included into a new section.

Au. A new section has been identified for the experimental evidence of the cardiovascular complications of diabetes as suggested (page 8, line 273).

  1. A perspective-related section is missing, before conclusions.

Au. A perspective section has been included in the revised manuscript as suggested (page 9, lines 310-340). Thank you.

  1. Line 233: how can the clinical evidences be strengthened? This ideas (and the subsequent ones) are very laconic. Authors can be more specific for readers.

Au. Additional comments related to the possibility to strengthen clinical use of oleuropein have been added in the revised version of the manuscript (page 9, lines 353-362).

  1. Revise the association of the manuscript content and the title, since the relationship of the benefits in the Mediterranean diet is only mentioned in the Introduction section. Perhaps in conclusions or perspectives sections something about this can be mentioned.

Au. The point is well taken. Accordingly, we have made additional reference to the Mediterranean diet in the new perspective section and in conclusions (page 9, lines 315-317, lines 319-320, lines 329-330, lines 343-346). Thank you.

Reviewer 2 Report

The article "The pivotal role of oleuropein in the anti-diabetic action of the Mediterranean Diet" presents a short overview on the role that polyphenol oleuroepein might play in the potential antidiabetic effects of the olive oil-rich Mediterranean diet. The article is soundly written. However, my main concern is that the Pharmaceutics does not publish commentaries as only the Original research articles and Reviews as the main article types that are published in the journal. In case the Commentaries can be accepted would recommend it for the publication provided that the following minor issues are fixed:

Section "Oleuropein and diabetes: clinical evidence", line 131 and further: Clinical trials are the most important types of the original scientific studies when considering efficacy of therapy for any disease. Furthermore, the results of these trials are most closely related to the existence of relevant antidiabetic effects of oleuropein which is the main topice of the article. Thus, please give more information on the clinical trials and the form of oleuropein used in them. What were the doses of oleuropein in the studies, what was the pharmaceutical dosage form, vehicle, duration of the studies, more details on the success of the therapy such as the percentage of the blood sugar levels reduction etc.
Besides, some minor, mostly typographycal and other errors may be fixed
Line 62, 64: in vitro, in vivo - please write in italic
Lines 64, 67,72 78, 79, 81, 135, 132/133, 209 and elsewhere: "-cells", "PPAR ", "-amylase" Greek characters are missing.
Lines 108,109 "(reviewed in reference 25)" is not necessary as the reference is given later in the text

Author Response

Reviewer 2

We thank the reviewer for his/her thoughtful comments and suggestions. Here are our responses to the points that were raised.

The article "The pivotal role of oleuropein in the anti-diabetic action of the Mediterranean Diet" presents a short overview on the role that polyphenol oleuroepein might play in the potential antidiabetic effects of the olive oil-rich Mediterranean diet. The article is soundly written. However, my main concern is that the Pharmaceutics does not publish commentaries as only the Original research articles and Reviews as the main article types that are published in the journal. In case the Commentaries can be accepted would recommend it for the publication provided that the following minor issues are fixed.

Au. Thank you. Submission of a commentary article was previously agreed with the Editorial office of Pharmaceutics upon their request. Upon request the has been changed to a minireview.

Section "Oleuropein and diabetes: clinical evidence", line 131 and further: Clinical trials are the most important types of the original scientific studies when considering efficacy of therapy for any disease. Furthermore, the results of these trials are most closely related to the existence of relevant antidiabetic effects of oleuropein which is the main topic of the article. Thus, please give more information on the clinical trials and the form of oleuropein used in them. What were the doses of oleuropein in the studies, what was the pharmaceutical dosage form, vehicle, duration of the studies, more details on the success of the therapy such as the percentage of the blood sugar levels reduction etc.

Au. The comment is appropriate, and we thank the Reviewer for giving us the opportunity to expand the revised manuscript with a table (Table 2) containing the details of the clinical studies conducted with oleuropein in diabetes. Also, we have added in text the doses of oleuropein that were used in the clinical trials (page 5, lines 191-192; page 6, lines 197 and 203; page 7, lines 216 and 240).

Besides, some minor, mostly typographical and other errors may be fixed
Line 62, 64: in vitro, in vivo - please write in italic
Lines 64, 67,72 78, 79, 81, 135, 132/133, 209 and elsewhere: "-cells", "PPAR ", "-amylase" Greek characters are missing.
Lines 108,109 "(reviewed in reference 25)" is not necessary as the reference is given later in the text

Au. Typographical errors have been corrected; Greek characters have been added and “(reviewed in reference 25)” has been removed. Thank you.

Reviewer 3 Report

In the manuscript: “The pivotal role of oleuropein in the anti-diabetic action of the 3

Mediterranean Diet”. The authors comment on the anti-diabetic properties of Oleuropein. An interesting topic that contributes to knowledge in the area, but certain issues must be corrected.

  1. In the introduction, the objective of the manuscript must be mentioned, mentioning what is the gap that the manuscript will fill within the current bibliography, the topics that will be reviewed in the manuscript and the possible conclusions that the reader will find through the manuscript.
  2. The manuscript must be reviewed by a native English speaker. Some sentences are vague and are not understood. For instance, the sentence in lines 63-65. In line 68 it is necessary to mention which are the metabolic pathways involved with PPAR.
  3. The sentences are very long, the authors must pause so that the reader can easily "digest" the information given in the manuscript. For example, this becomes evident in lines 74-76, where there are no pauses in what is desired say and, in the end, the reader loses the meaning of the sentences. The sentence on lines 88-93 has the same problem.
  4. In line 87 it is necessary to mention the role of oxidative stress during the disease or mention if oxidative stress is a relevant characteristic in this model.
  5. Authors should make use of a table that easily guides the reader on the properties of oleuropein. Likewise, the authors should produce figures on the oleuropein mechanisms that would contribute greatly to the fulfillment of the objective of the manuscript.
  6. The authors must mention what restoration of histopathological characteristics they refer to in line 100.
  7. Authors are encouraged to include in each section or paragraph a conclusion from the data reviewed.
  8. The authors must mention the pieces of evidence that support that oleuropein reduces oxidative stress, inflammation, and cell death; that is, they have to mention which are the markers that confirm these effects.
  9. In line 214 the authors must mention how nitric oxide is an antioxidant since it has been reported that this molecule can react with the superoxide radical producing peroxynitrite, a highly reactive oxygen species.

Author Response

Reviewer 3

We thank the reviewer for his/her thoughtful comments and suggestions. Here are our responses to the points that were raised.

In the manuscript: “The pivotal role of oleuropein in the anti-diabetic action of the 3 Mediterranean Diet”. The authors comment on the anti-diabetic properties of Oleuropein. An interesting topic that contributes to knowledge in the area, but certain issues must be corrected.

Au. Thank you.

  1. In the introduction, the objective of the manuscript must be mentioned, mentioning what is the gap that the manuscript will fill within the current bibliography, the topics that will be reviewed in the manuscript and the possible conclusions that the reader will find through the manuscript.

Au. Additional text clarifying the main purpose of the manuscript, the covered topics, and the overall information provided to the readership has been added to the revised version of the manuscript as suggested (page 2, lines 65-75). Thank you.

  1. The manuscript must be reviewed by a native English speaker. Some sentences are vague and are not understood. For instance, the sentence in lines 63-65. In line 68 it is necessary to mention which are the metabolic pathways involved with PPAR.

Au. As requested, the manuscript has been reviewed by a native English speaker and hopefully text is now more fluent and understandable. The sentence in lines 63-65 has been changed. The metabolic pathways related to PPARa activation have been mentioned in the revised manuscript (page 3, lines 106-110). Thank you.

  1. The sentences are very long, the authors must pause so that the reader can easily "digest" the information given in the manuscript. For example, this becomes evident in lines 74-76, where there are no pauses in what is desired say and, in the end, the reader loses the meaning of the sentences. The sentence on lines 88-93 has the same problem.

Au. Many sentences in text have been rephrased and shortened according to the Reviewer’s suggestion to make them easily understandable. Thank you.

  1. In line 87 it is necessary to mention the role of oxidative stress during the disease or mention if oxidative stress is a relevant characteristic in this model.

Au. Oxidative stress is a relevant characteristic in the model of alloxan-induced diabetes since a significant rise in plasma and erythrocyte malondialdehyde and changes in enzymatic and non-enzymatic antioxidants is observed. This has been clarified in the revised manuscript (page 4, lines 128-129).

  1. Authors should make use of a table that easily guides the reader on the properties of oleuropein. Likewise, the authors should produce figures on the oleuropein mechanisms that would contribute greatly to the fulfillment of the objective of the manuscript.

Au. A new table (Table 1) summarizing the properties of oleuropein and a new figure (Figure 1) with the chemical structure of oleuropein and the mechanisms that are involved in regulation of glucose homeostasis and insulin sensitivity have been included in the revised version of the manuscript.

  1. The authors must mention what restoration of histopathological characteristics they refer to in line 100.

Au. The db/db mice model is characterized by pancreatic islet damage, lipid deposition in hepatocytes, and disordered myocardial fibers. These histopathological changes were restored by oleuropein administration. This information has been added in the revised version of the manuscript (page 4, lines 140-143).

  1. Authors are encouraged to include in each section or paragraph a conclusion from the data reviewed.

Au. A summary statement has been included at the end of each paragraph as requested (page 4, lines 161-164; page 7, lines 233-237; page 8, lines 268-271; page 8, lines 306-308). Thank you.

  1. The authors must mention the pieces of evidence that support that oleuropein reduces oxidative stress, inflammation, and cell death; that is, they have to mention which are the markers that confirm these effects.

Au. Evidence linking oleuropein to reduced oxidative stress, inflammation, and cell proliferation has been added to the revised version of the manuscript together with specific references (page 2, lines 82-84; Table 1; references 12,13,14) as requested. Additional mention of specific markers of oxidative stress, inflammation, and apoptosis that are related to glucose balance has been included throughout text (page 4, lines 128-129; page 5, lines 177-178; page 7, lines 221-222 and 248-249; page 8, lines 256-259, lines 261-264, lines 293-295, lines 298-299).

  1. In line 214 the authors must mention how nitric oxide is an antioxidant since it has been reported that this molecule can react with the superoxide radical producing peroxynitrite, a highly reactive oxygen species.

Au. The point is well taken, and this is an important issue. Nitric oxide generated by inducible nitric oxide synthase (iNOS) can be the substrate for formation of peroxynitrite and thereby contribute to increase oxidative stress. However, nitric oxide generated by the regulated endothelial nitric oxide synthase (eNOS) has antioxidant and anti-inflammatory effects together with multiple additional vasoprotective and metabolic properties. At physiological levels, nitric oxide acts as an antioxidant blocking fenton-type reactions, terminating radical chain reactions, and inhibiting peroxidases and oxidases (Forstermann et al. Circ Res 2017,120:713-735) thus decreasing oxidative stress. This has been clarified in the revised manuscript (page 8, lines 293-295).

Reviewer 4 Report

The authors have described the benefit of oleuropein intake and its anti-diabetic effects. The review is nicely written but precisions are lacking at sevral places

1- a comment should be made on doses. How much oil should be consumed for any benefit? it is mandatory that doses should be introduced

2- a table summarizing the different studies reporting dose levels of oleuropein is required. Too much of text but no figure and no table

3- the authors do not precise if oleuropein has preventive effects or if it can alleviate metabolic disorders? or both?

4- line 72: PPARg has insulin sensitizing properties which do not fit with the overall sentence line 72 of the review.

5- analysis of the microbiota is now very well developed and a lot is known on the different phylla, on SCFAs and the beneficial phylla. Therefore, it is not sufficient to write about beneficial effects without further explaining what is happening? which phylla are modified etc

6- lines 163-170: the paragraph may lead to the conclusion that chocolate is good as long as you incorporate oleuropein. Please modify the text

7- all symbols have disappeared from the pdf text. Please reintroduce them

Author Response

Reviewer 4

We thank the reviewer for his/her thoughtful comments and suggestions. Here are our responses to the points that were raised.

The authors have described the benefit of oleuropein intake and its anti-diabetic effects. The review is nicely written but precisions are lacking at several places.

Au. Thank you.

1- a comment should be made on doses. How much oil should be consumed for any benefit? it is mandatory that doses should be introduced

Au. This is an important point. Doses of oleuropein that were used in clinical trials are now shown in text (page 5, lines 191-192; page 6, lines 197 and 203; page 7, lines 216 and 240) and are summarized in a new table (Table 2). Thank you.

2- a table summarizing the different studies reporting dose levels of oleuropein is required. Too much of text but no figure and no table

Au. A table (Table 2) containing the details of the clinical studies conducted with oleuropein in diabetes has been added in the revised version of the manuscript.

3- the authors do not precise if oleuropein has preventive effects or if it can alleviate metabolic disorders? or both?

Au. Current evidence suggests that oleuropein could provide both preventive and therapeutic benefits in type 2 diabetes.  We have clarified this point in the revised version of the manuscript (page 1, lines 22-23; page 7, lines 236-237; page 9, lines 339-340). Thank you.

4- line 72: PPARg has insulin sensitizing properties which do not fit with the overall sentence line 72 of the review.

Au. The point is correct. PPARg stimulation increases insulin sensitivity but also promotes intracellular fatty acids deposition. That specific sentence refers to adipose tissue mass and liver fat deposits. This has been clarified in the revised version of the manuscript (page 3, lines 110-114) highlighting the relevance that PPARg stimulation might have for the development of obesity. Thank you.

Reviewer 5 Report

The authors have submitted a hypothesis article, proposing that oleuropein may play an important role in the antidiabetic effects of the Mediterranean Diet.

The manuscript is interesting but I am not sure if Pharmaceutics accepts commentary submissions (I could not find this in the Instructions for Authors). I would recommend the authors to transform the paper into a (mini)review. It would be interesting to see a systematic review on the antidiabetic actions of oleuropein (for example, PubMed/MEDLINE:  oleuropein AND (diabetes OR diabetic OR antidiabetic)).

Please adjust the template of the paper and add an abstract.

Add a methodology to explain how the inclusion/exclusion of articles included in the discussion was performed.

It would be interesting to see the chemical structures of the discussed compound(s).

In addition, the authors could discuss other bioactive molecules with antidiabetic effects, as well as other diets that have antidiabetic/glucose-lowering effects. See:

https://pubmed.ncbi.nlm.nih.gov/33253361/

https://pubmed.ncbi.nlm.nih.gov/33492173/

https://pubmed.ncbi.nlm.nih.gov/33966619/

Revise the references to match the style of the journal (American Chemical Society).

Moreover, it should be taken into account that the products recommended in the Mediterranean diet contain a myriad of bioactive compounds that have a multitude of health benefits. 

Author Response

Reviewer 5

We thank the reviewer for his/her thoughtful comments and suggestions. Here are our responses to the points that were raised.

The authors have submitted a hypothesis article, proposing that oleuropein may play an important role in the antidiabetic effects of the Mediterranean Diet.

1- The manuscript is interesting, but I am not sure if Pharmaceutics accepts commentary submissions (I could not find this in the Instructions for Authors). I would recommend the authors to transform the paper into a (mini)review. It would be interesting to see a systematic review on the antidiabetic actions of oleuropein (for example, PubMed/MEDLINE:  oleuropein AND (diabetes OR diabetic OR antidiabetic)).

Please adjust the template of the paper and add an abstract.

Add a methodology to explain how the inclusion/exclusion of articles included in the discussion was performed.

Au. Thank you. Submission of a commentary article was previously agreed with the Editorial office of Pharmaceutics upon their request. The article in fact turned out to have the size of a minireview and an abstract (page 1) and the methodology for collection of articles (page 2, lines 70-71) has been added in the revised version of the manuscript as suggested. Thank you.

2- It would be interesting to see the chemical structures of the discussed compound(s).

Au. A figure (Figure 1) with the chemical structure of oleuropein has been added together with description of the mechanisms that are involved in regulation of glucose balance.

3- In addition, the authors could discuss other bioactive molecules with antidiabetic effects, as well as other diets that have antidiabetic/glucose-lowering effects. See:

https://pubmed.ncbi.nlm.nih.gov/33253361/

https://pubmed.ncbi.nlm.nih.gov/33492173/

https://pubmed.ncbi.nlm.nih.gov/33966619/

Moreover, it should be taken into account that the products recommended in the Mediterranean diet contain a myriad of bioactive compounds that have a multitude of health benefits. 

Au. This is an important point although the article was meant to be focused just on oleuropein. Additional comments on other bioactive molecules with antidiabetic effects have been added in the revised manuscript as suggested (page 9, lines 327-330). Also, reference to the myriads of bioactive compounds that are present in the Mediterranean diet has been emphasized in the revised manuscript (page 9, lines 319-327). References 44 and 45 have been added. Thank you.

4- Revise the references to match the style of the journal (American Chemical Society).

Au. The style of references has been revised to match the style of the journal. Thank you.

5- analysis of the microbiota is now very well developed and a lot is known on the different phylla, on SCFAs and the beneficial phylla. Therefore, it is not sufficient to write about beneficial effects without further explaining what is happening? which phylla are modified etc

Au. The comment is appropriate and additional comments on the influence of intestinal bacteria in regulation of metabolic pathways have been added in the revised manuscript (page 4, lines 154-160). Also, details of phylla that were modified in animals who were treated with oleuropein have been added in text (page 4, lines 145-148). Thank you.

6- lines 163-170: the paragraph may lead to the conclusion that chocolate is good as long as you incorporate oleuropein. Please modify the text

Au. The sentence has been modified as requested (page 7, lines 229-232). Thank you.

7- all symbols have disappeared from the pdf text. Please reintroduce them

Au. Unfortunately, symbols were removed by the word processor and have been reintroduced. Thank you.

Round 2

Reviewer 3 Report

The authors must complete Table 1 with the references of each work.

The authors must describe Figure 1 in the figure legend.

The authors must define all abbreviations (line 94: GLUT, line 118: INS-1, etc.)

Confirm that "nitric oxide synthase (eNOS) functions as an antioxidant" (line 293), since NO production can react with the superoxide radical to produce ONOO-, a highly reactive oxygen species (doi: 10.1152/physrev.00029.2006

Author Response

Reviewer 3

We thank the Reviewer for his/her further comments and suggestions. Here are our responses.

The authors must complete Table 1 with the references of each work.

Au. Table 1 has been completed with the appropriate references cited in text.

The authors must describe Figure 1 in the figure legend.

Au. Figure 1 has been described in an additional text of the legend (page 5, lines 178-184).

The authors must define all abbreviations (line 94: GLUT, line 118: INS-1, etc.).

Au. Abbreviations have been defined.

Confirm that "nitric oxide synthase (eNOS) functions as an antioxidant" (line 293), since NO production can react with the superoxide radical to produce ONOO-, a highly reactive oxygen species (doi: 10.1152/physrev.00029.2006).

Au. Thank you for the opportunity to clarify this important issue. Nitric oxide can have different effects under physiological and pathological conditions. As reported by Pacher P. et al. (Physiol Rev. 87:315-424, 2007; doi: 10.1152/physrev.00029.2006; legend to figure 9, page 162) nitric oxide by activating soluble guanylate cyclase signal transduction pathway mediates various physiological/beneficial effects including vasodilation, inhibition of platelet aggregation, anti-inflammatory, antiremodelling, and antiapoptotic effects. On the other hand, under pathological conditions associated with increased oxidative stress and inflammation, NO and superoxide (O2•−) react to form peroxynitrite (ONOO−) which induces cell damage via lipid peroxidation. This has been further clarified in the revised manuscript (page 9, lines 315-317).

Reviewer 4 Report

Tables: the authors should indicate the references which have been used for making the Table

Doses are only indicated in clinical investigations. It should also be stated for experimental studies and a comparison should be made; also the authors should discuss doses versus the diet intake of oil

Author Response

Reviewer 4

We thank the Reviewer for his/her further comments and suggestions. Here are our responses.

Tables: the authors should indicate the references which have been used for making the Table

Au. Table 1 has been completed with the appropriate references cited in text.

Doses are only indicated in clinical investigations. It should also be stated for experimental studies and a comparison should be made; also, the authors should discuss doses versus the diet intake of oil.

Au. Doses used in the experimental studies have been reported in the revised manuscript. Also, comments on these doses (page 4, lines 167-170) and on the  and doses of oleuropein versus the diet intake of oil have been added (page 7, lines 255-257).

Reviewer 5 Report

Well-done! I was unaware that the journal accepted commentaries and that it was agreed upon for you to submit this manuscript. However, I think that the paper has been significantly improved and that the mini-review format suits the subject better. In this revised form, I strongly believe the paper is suitable for publication and I commend the work of the authors.

Author Response

Reviewer 5

Well-done! I was unaware that the journal accepted commentaries and that it was agreed upon for you to submit this manuscript. However, I think that the paper has been significantly improved and that the mini-review format suits the subject better. In this revised form, I strongly believe the paper is suitable for publication and I commend the work of the authors.

Au. Thank you.